# Application of Ethyl Cellulose and Ethyl Cellulose + Polyethylene Glycol for the Development of Polymer-Based Formulations using Spray-Drying Technology for Retinoic Acid Encapsulation

**DOI:** 10.3390/foods11162533

**Published:** 2022-08-22

**Authors:** Antónia Gonçalves, Fernando Rocha, Berta N. Estevinho

**Affiliations:** 1LEPABE—Laboratory for Process Engineering, Environment, Biotechnology and Energy, Department of Chemical Engineering, Faculty of Engineering, University of Porto, Rua Dr. Roberto Frias, 4200-465 Porto, Portugal; 2ALiCE—Associate Laboratory in Chemical Engineering, Faculty of Engineering, University of Porto, Rua Dr. Roberto Frias, 4200-465 Porto, Portugal

**Keywords:** controlled delivery, ethyl cellulose, polyethylene glycol, release models, retinoic acid, spray-drying

## Abstract

Ethyl cellulose (EC)-based microparticles, with and without the incorporation of polyethylene glycol (PEG) as a second encapsulating agent, were prepared using the spray-drying process for the encapsulation of retinoic acid (RA). The production of a suitable controlled delivery system for this retinoid will promote its antitumor efficiency against acute promyelocytic leukemia (APL) due to the possibility of increasing the bioavailability of RA. Product yield ranged from 12 to 28% in all the microparticle formulations, including unloaded microparticles and RA-loaded microparticles. Microparticles with a mean diameter between 0.090 ± 0.002 and 0.54 ± 0.02 µm (number size distribution) and with an irregular form and rough surface were obtained. Furthermore, regarding RA-loaded microparticles, both polymer-based formulations exhibited an encapsulation efficiency of around 100%. A rapid and complete RA release was reached in 40 min from EC− and EC + PEG-based microparticles.

## 1. Introduction

Retinoic acid (RA) is a natural derivative of vitamin A considered a chemopreventive and/or chemotherapeutic agent [1]. To treat several malignancies, this signaling molecule is involved in inhibiting cellular proliferation, migration and invasion, and the triggering of differentiation, apoptosis and cellular cycle arrest in pre-malignant and malignant cells. These processes are carried out by gene transcription regulation due to RA binding predominantly to RARs (RA receptors) [2,3,4,5,6,7].

Despite the high efficacy of all-trans RA to treat acute promyelocytic leukemia (APL) (induction agent), the concentration of this retinoid that reaches and is therefore maintained in the target cells and tissues may not be in the required amount to activate target genes [8]. Consequently, therapeutic efficiency is compromised with patients becoming resistant to RA. Moreover, due to continuous repetitive RA oral administration, a high decline in the RA half-life in blood and a further reduction in the concentration of this retinoid in the plasma of patients is reported [9,10,11,12]. Muindi et al. [11] presented some hypotheses for this reduction of RA half-life, including (1) a marked elimination of RA by biliary excretion (at least 60%); herein, nonspecific binding to the intestinal proteins or unreal malabsorption decreased oral bioavailability of RA, (2) RAR-α mutations and (3) the induction of RA catabolism due to the continued treatment with RA. Considering this last possibility, the regulation of cellular levels of RA is related to cytochrome P450 enzymes. They are activated for low amounts of this retinoid, whereby repeated RA administration may stimulate RA degradation with the formation of 4-oxo-all-trans RA [1,13].

In line with these topics, incorporating RA into proper carrier-based delivery systems for its slow and continuous release is of major importance to boost the treatment of APL. Repetitive RA administration is herein avoided, which decreases the activity of cytochrome P450 enzymes and thus improves the bioavailability of RA. This approach also may help to deal with inactivity or rapid degradation of RA due to its poor aqueous solubility and high physicochemical instability (e.g., under oxidant and heat conditions). Also, minimization of RA side-effects due to the high amount of RA administrated might be achieved [14].

The development of polymeric microparticles using spray-drying for RA encapsulation has been proposed by the authors as a strategy addressed for patients with APL. Gonçalves et al. [15] evaluated the incorporation of RA (ethanolic solution) into modified chitosan-, alginic acid sodium- and arabic gum-based microparticles. Alginic acid sodium-based microparticles showed a RA release for more than 7 h. Afterward, oil-in-water emulsions with RA (solutions prepared in coconut oil), tween 80 and biopolymers (individually used [16] and combined in advantageous binary and ternary blends [17]) resulted in RA-loaded microparticles with variable characteristics. The higher encapsulation efficiency was registered for alginic acid sodium-, modified chitosan- and xanthan gum + alginic acid sodium-based microparticles (62, 65, and 76%, respectively). Also, RA controlled release was prolonged for almost 8 h for alginic acid sodium-based microparticles, and for almost 7 h for the majority of microparticles with alginic acid sodium in the biopolymers blends (the exception refers to microparticles composed by the modified chitosan + alginic acid sodium blend, wherein RA release was carried out in 100 min).

Spray-drying is a one-step procedure that enables the continuous (continuous batch), rapid and reproducible production of microparticles as carrier-based delivery systems at a low cost. It is also an easy scale-up process and enables the production of good-quality microparticles [18]. In the present work, two synthetic polymers—ethyl cellulose (EC) and polyethylene glycol (PEG)—were considered to design microparticle formulations for RA oral administration and provide its controlled delivery in the intestine. These materials are hypothesized as an approach to prepare carrier-based systems to be fed to the spray-dryer with a significantly higher mass of RA when compared to the maximum mass of RA that was possible to add to the biopolymer-based systems previously designed by the authors [15,16,17]. Moreover, the possibility of increasing the encapsulation efficiency using synthetic polymer-based microparticles was also considered. This step is of major importance to ensure that the microparticles produced have a suitable amount of RA for treating APL. The usual daily dose of Vesanoid^®^ is 80 mg for one adult (patient information leaflet). EC-based microparticles and microparticles composed by the EC + PEG blend were produced. EC is a cellulose-derived polymer with biocompatible, non-allergic, non-irritant, tasteless and non-biodegradable properties. It is not water-soluble but soluble in several organic solvents (e.g., alcohols). Moreover, EC is described as a lipophilic polymer with a great ability to form membranes and films with good thermostability and durability. EC is generally recognized as safe (GRAS) and has been widely explored in the pharmaceutical industry to develop a drug carrier to prepare microparticles capable of providing a sustained release. In fact, EC has been described as a retardant of drug release despite the medium’s acidity and is therefore used in the composition of oral delivery systems [19,20]. In turn, PEG is biocompatible, non-toxic, presents water solubility and is used in pharmaceutical and food formulations. Accordingly, the PEG inclusion in an EC formulation is carried out to increase the drug release rate due to the solubilization of PEG in the gastrointestinal tract. The concentration of EC and PEG can be used to manage and reach the desired release rate [21].

## 2. Materials and Methods

### 2.1. Materials

All-trans RA (≥98%, R2625-1G, CAS 302-79-4) and EC tested according to Ph. Eur. (28244-250G, Lot#BCBQ4872V, CAS 9004-57-3, viscosity: 10 mPa.s, ethoxy groups (dried substances): 48.0–49.5%, country of manufacture: Saint Louis, MO, USA) were obtained from Sigma-Aldrich. PEG 4000 for synthesis (807490-1KG, LOT#5191687, average molecular mass: 3500–4500) and 1-octanol for synthesis (820931, CAS 111-87-5) were supplied by Merck (German). Absolute ethanol (83813.360, Lot 21B104029, CAS 64-17-5) was acquired from VWR BDH Chemicals.

### 2.2. Formulation of RA Loaded-Polymer-Based Microparticles by Spray-Drying Technology

EC was used individually (EC, 100%) and as a blend of materials (EC + PEG, 50–50%) to prepare solutions in absolute ethanol (1% (*w*/*v*)). RA was combined with the polymer-based solutions 24 h after their preparation (after full hydration of polymers) in a final concentration of 0.1% (*w*/*v*). Each solution was prepared at 35 ± 2 °C and under stirring conditions (500 rpm) and instantly fed to the spray-dryer. A mini spray-dryer B-290 from BÜCHI (Flawil, Switzerland) with a standard cyclone and a standard 0.5 mm nozzle, coupled to an Inert Loop B-295 from BÜCHI (Flawil, Switzerland), was used to produce microparticles for RA encapsulation. Nitrogen pressure, aspiration rate and solution feed flow were set at 2 bar, 36 m^3^∙h^−1^ (100%) and 4 mL∙min^−1^, respectively. Inlet air temperature was set at 95 °C and outlet temperature varied between 41 and 51 °C. The temperature of Inert Loop B-295 was set to −5 °C. The microparticles produced were collected from the cyclone and the collection vessel of the spray-dryer. Then the dry powder was covered with aluminum foil and stored under refrigeration. The product yield (%) was calculated based on Equation (1).
(1)$$Product\;yield\;\left(\%\right)=\frac{Dry\;powder\;weight\;recovered\;in\;the\;cyclone\;and\;in\;the\;collection\;vessel\;ofspray-dryer\;}{Total\;mass\;of\;materials\;used\;to\;prepare\;the\;systems\;fed\;to\;the\;spray-dryer\;}\times 100$$

### 2.3. Microparticle Morphology

A Fei Quanta 400 FEG ESEM/EDAX Pegasus X4M (Eindhoven, The Netherlands) was used to assess microparticles morphology by scanning electron microscopy. Herein, microparticles were placed on a brass stub with double-sided adhesive tape and subsequently coated by a gold-palladium thin layer using an SPI module sputter coater (vacuum conditions).

### 2.4. Particle Size Distribution

The mean diameter of microparticles was measured using a Coulter LS 230 Particle Size Analyser (Coulter—Miami, FL, USA) utilizing laser granulometry. A suspension of microparticles was prepared in water and analyzed regarding volume and number distributions. The volume of each particle is considered in volume distribution, whereby a greater contribution is attributed to particles with a higher volume. In turn, each particle has an equal weight in the number distribution. Measurements were performed in triplicate, each with a duration of 30 s.

### 2.5. Encapsulation Efficiency and Loading Capacity

Encapsulation efficiency (%) was evaluated using the ratio between the mass of RA evidenced in the microparticles and the mass of RA added to the polymer-based solutions prior to their feeding to the spray-dryer, based on the approach implemented by Gonçalves et al. [16,17] and adapted from Côrrea-Filho et al. [22]. RA-loaded microparticles were suspended in octanol (analysis in triplicate), stirred (2000 rpm, 2 min) and maintained on the laboratory bench overnight. Then the suspension was stirred again (2000 rpm, 5 min), placed on a bath (37 ± 2 °C, 3 h) and centrifuged. The quantification of RA in the supernatant was assessed by a UV/VIS spectrometer (SPEC RES+, Sarspec, Porto, Portugal) at 352 ± 7 nm. The calibration curve was validated between 0.00008 and 0.006 mg∙mL^−1^ (correlation coefficient of 0.9927). LOD and LOQ were 0.0002 mg∙mL^−1^ and 0.0007 mg∙mL^−1^, respectively.

The ratio between the mass of RA and the mass of microparticles used (analysis in triplicate) enabled to determine the loading capacity.

### 2.6. Controlled Release Experiments

Microparticles loaded with RA were put on a beaker with octanol (an organic solvent capable of imitating the characteristics of phospholipids membrane, considering its amphiphilic nature) [23,24,25,26] under magnetic stirring conditions and at 37 ± 2 °C. At appropriate time intervals, samples were recovered, analyzed in a UV/VIS spectrometer (Section 2.5) by absorbance (at 350 nm) and returned to the beaker. With the data obtained, RA release profiles from each type of microparticles were drawn. The models of zero-order (Equation (2)), Higuchi (Equation (3)), Korsmeyer-Peppas (Equation (4)) and Weibull (Equation (5)) [27,28,29,30] were fitted to the obtained controlled release profiles.
(2)$$Q_{t}\;=\;Q_{0}+K_{0}t$$
where *Q_t_* is the cumulative amount of the bioactive compound released at time *t*, *Q*_0_ is the initial amount of the bioactive compound in solution and *K*_0_ is the zero-order release constant.
(3)$$Q_{t}\;=\;K_{H}\;\sqrt{t}$$
where *K_H_* is the Higuchi constant.
(4)QtQ∞ = KK tn
where *Q_t_*_⁄_*Q_∞_* is the portion of bioactive compound released until time *t*, *K_K_* is the Korsmeyer constant and *n* is the release exponent (diffusional) and indicates the release mechanism.
(5)$$M_{t}=M_{\infty }\left[1-{e^{-\left(\frac{t-t_{0}}{\tau_{d}}\right)}}^{\beta }\right]$$
where *M_t_* is the release (%) of the bioactive compound at time *t*, *M_∞_* is the release (%) of the bioactive compound at infinite time, *t*_0_ is the lag-time of the release, *β* points out the shape parameter of the curve, and *τ_d_* is the time when 63.2% of *M* has been released.

### 2.7. Statistical Analysis

The particle size of microparticles was evaluated for all the microparticle formulations. Moreover, encapsulation efficiency and loading capacity were analyzed for RA-loaded microparticles with different polymers in their composition. One-way analysis of variance (ANOVA) at *p* < 0.05 and consequent Tukey test were performed in Minitab Statistical Software (license provided from The Department of Industrial Engineering and Management of the Faculty of Engineering of the University of Porto to Antónia Gonçalves).

## 3. Results and Discussion

EC-based microparticles and microparticles composed by the EC + PEG blend were designed as RA-controlled delivery systems regarding the spray-drying process. PEG-based microparticles were also produced. However, the solution composed by RA and PEG dissolved in absolute ethanol fed to the spray-dryer resulted in microparticles with a crystalline appearance that remained adhered to the cyclone of the spray-dryer. Briefly, this can be due to the PEG responses to the heating and cooling rate during spray-drying operating conditions. The behavior of PEG crystallization can change with thermal cycles. Accordingly, RA-loaded PEG microparticles were not possible to be recovered. The product yield of unloaded microparticles (microparticles MpPE—only composed by the polymer[s] as encapsulating agent[s]) and RA-loaded microparticles (microparticles MpPRA) changed between 12 and 28% (Table 1). This parameter is mostly affected by the formulation (i.e., solution, suspension, or emulsion) fed to the spray-dryer and the properties of the materials used as encapsulating agents. Moreover, the volume of the formulation prepared and the conditions used to operate the spray-drying technique also influence the product yield. In this work, a spray-dryer was operated based on conditions previously used by some of the authors [31]. All the solutions were prepared with 100 mL, varying their composition in terms of the polymers used and the presence/absence of bioactive compounds. Using EC as an individual encapsulating agent or in a blend with PEG resulted in a similar product yield among microparticles MpPE (Table 1).

In turn, incorporating RA in the microparticles decreased the product yield of EC-based microparticles while increased the product yield of EC + PEG-based microparticles (Table 1). The type of polymers used seems to not interfere with product yield. However, the impact of the encapsulating agents’ properties on the product yield is made clear in several studies. For example, the high viscosity of xanthan gum requires the preparation of a higher volume of solutions/emulsions based on this polysaccharide when compared to the one used to prepare solutions/emulsions composed by low viscosity biopolymers to obtain a similar product yield [16]. Furthermore, the product yield of polymer-based microparticles obtained in this work was lower when compared to the biopolymer-based microparticles produced in other works using the same encapsulating method [32,33,34]. Herein, a great amount of EC− and EC + PEG-based microparticles remained deposited on the cyclone of the spray-dryer, which compromised the recovery of the dried powder. Several authors report the production of EC-based microparticles with a product yield typically higher than 40%. Waghulde and Naik [35] produced EC microparticles using the spray-drying technique for vildagliptin (an antihyperglycemic drug) encapsulation with a product yield between 40.85 and 75.73%. This experiment was carried out considering a two-factor three-level factorial design wherein the surfactant concentration (span-80, between 0.3 and 0.8% *w*/*v*) and mass of EC (between 0.5 and 1.5 g) were selected as independent variables. EC solutions were prepared in methanol. The obtained particles exhibited a mean size of 1.026 µm. Also, Stocker et al. [36] and Tsolaki et al. [37] spray-dried active pharmaceutical ingredient-ionic liquids into EC with a product yield that ranged between 67.9 and 87.8% (only in two cases was the product yield 24.8% and 29.9%), using a mini spray-dryer combined with the B-295 inert loop and two-fluid nozzle (1.5 mm cap and 0.7 mm tip). The EC concentration used was 2.5% (*w*/*v*).

### 3.1. Surface Morphology of Microparticles

Microparticles MpPE and MpPRA, both composed by EC and by the EC + PEG blend, exhibited an irregular form that varied between a spherical and elliptic structure (Figure 1). Moreover, they evidenced several concavities that determined a rough surface. In this case, the production of microparticles with different formulations did not result in different microparticle morphologies. The polymers used as encapsulating agents can influence the morphology of microparticles produced. Moreover, it can be influenced by the spray-drying process. Waghulde and Naik [35] also evaluated the morphology of microparticles produced, being very similar to the one registered for the microparticles obtained in this work. On the other hand, Wagh et al. [38] obtained by spray-drying EC micro-/nanospheres loaded with ketorolac tromethamine that showed a spherical form and a slightly rough surface. In the preparation of micro-/nanospheres the drug and polymer were dissolved in a mixture of dichloromethane and acetone along with magnesium stearate.

### 3.2. Microparticle Particle Size

Mean diameter of all microparticle formulations ranged between 0.090 ± 0.002 and 0.54 ± 0.02 µm (Table 1) (differential number distribution). In other studies, using spray-drying technology to produce microparticles composed by different biopolymers (individual or as a blend) for the controlled delivery of several bioactive compounds resulted in a powder with a mean size greater than 0.100 µm [15,39,40]. In turn, differential volume distribution showed a general particle mean diameter between 6.0 ± 0.1 and 44 ± 8 μm, the smaller size related to microparticles MpPE composed by the EC + PEG blend (Table 1). These results suggest the aggregation of microparticles, which was identified on the volume weighted size distribution (Figure 2). This explains the marked divergence of particle mean diameter exhibited comparing volume and number distribution. Waghulde and Naik [35] obtained EC microparticles by spray-drying with a mean size of 1.026 µm, as previously described (Section 3). Volume weighted mean diameter (D (4,3)) varied from 6.0 ± 0.1 and 44 ± 8 μm and area-volume mean diameter (D (3,2)) varied from 1.97 ± 0.03 and 6.1 ± 0.4 μm (Table 1). Lastly, the span of the volume-based distribution ranged from 3.0 ± 0.3 and 7 ± 3 μm (Table 1).

### 3.3. Encapsulation Efficiency and Loading Capacity

Encapsulation efficiency was experimentally calculated as 107 ± 23 and 109 ± 19% for EC− and EC + PEG- based microparticles, while loading capacity was 10 ± 2% for both types of microparticles (Table 1). In this work, the production of polymer-based delivery microparticles capable of incorporating the total mass of RA used to formulate the systems fed to the spray-dryer is a major achievement. It must be pointed out that the usage of an increased amount of RA is of a suitable amount for treating APL. With an encapsulation efficiency of 100%, the amount of EC− and EC + PEG- based microparticles capable of incorporating the current daily amount of RA administrated for APL is less than 1 g.

In previous studies, biopolymer-based microparticles produced by spray-drying technology for RA encapsulation were designed by mixing a RA solution (prepared in coconut oil at 0.1% (*w*/*v*)), tween 80 (0.5% (*w*/*v*) regarding RA solution) and each encapsulating agent solution. Herein, the higher encapsulation efficiency was 65 ± 6% using modified chitosan-based microparticles [16] and 76 ± 4% for microparticles composed by the xanthan gum + alginic acid sodium blend [17].

The encapsulating agents used in the microparticle production highly influence the bioactive compound content effectively retained in the carrier-based systems [41]. Waghulde and Naik [35] registered an encapsulation efficiency between 71.42 and 89.87% for the vildagliptin-loaded microparticles composed by EC. In turn, Wagh et al. [38] observed an encapsulation efficiency between 70.74 and 79.68% for the ketorolac tromethamine-loaded micro-/nanospheres also composed by EC.

### 3.4. Drug Controlled Release

RA release from EC-based microparticles and microparticles composed by the EC + PEG blend was performed at 37 °C in octanol. This solvent will enable to assume RA’s ability to cross the lipid membranes [23,24,25]. Carrier-based delivery systems composed by EC are resistant to acidic environments and prolong the drug release for a long time, as previously referred to (Section 1) [19,20]. In turn, the EC + PEG blend arises as a strategy to increase the drug release due to PEG solubilization [21]. Accordingly, in this study, it was assumed that the microparticles produced are not significantly damaged until they reach the intestine, wherein the release and absorption of RA are carried out in contact with the lumen membrane.

The obtained cumulative RA release profiles are presented in Figure 3. The release of RA behavior from both types of microparticles was very similar and is characterized by a fast release of this drug into the surrounding environment. Complete RA release was achieved in 40 min when the polymer EC and the EC + PEG blend were considered encapsulating agents. Several studies report the influence of different controlled release conditions on the drugs’ release rate from EC-based microparticles. For example, Waghulde and Naik [35] registered a continuous and constant release rate of vildagliptin from spray-dried EC microparticles in a phosphate buffer (pH 7.4) up to 12 h. At the end of 12 h, between 89.45 and 97.4% of the drug was released. Wagh et al. [38] also used a phosphate buffer (pH 6.8) to investigate the release of ketorolac tromethamine from EC microparticles obtained by spray-drying. Herein, drug release varied from 32.68 to 45.48% during the 12 h of the experiment. de Francisco et al. [42] evaluated the fluoride release from EC microparticles and a fast release of sodium fluoride, sodium monofluorophosphate, and aminofluoride in purified water was evidenced in the first 30 min.

### 3.5. Kinetic Models

Kinetics of zero order were adjusted to the controlled release profiles of RA obtained from EC− and from EC + PEG-based microparticles with a determination coefficient of 0.774 and 0.751, respectively (Table 2). The zero-order release constant, *K*_0_, was 7.1 × 10^−3^ and 7.2 × 10^−3^ mg∙min^−1^, suggesting a similar release rate of RA from both types of microparticles.

Regarding the Higuchi model, the Higuchi constant, *K_H_*, registered the values 3.6 × 10^−2^ and 3.7 × 10^−2^ mg∙min^−0.5^ when the polymer EC and the EC + PEG blend were used in the microparticle formulations, respectively. Furthermore, the determination coefficient were 0.977 and 0.972, respectively.

A *Q_t_⁄Q*_∞_ lower than 0.6 was considered to fit the Korsmeyer-Peppas model to the RA-controlled release profiles [29]. For all the microparticles produced, the parameter *n* was lower than 0.43 (*n* = 0.024 for EC-based microparticles and *n* = 0.0040 for the microparticles composed by the EC + PEG blend [Table 2]), whereby a Fickian diffusion (case I transport) and a rate as a function of time of *t*^−0.57^ is suggested [29,43]. The Korsmeyer constant, *K_K_*, and the determination coefficients were 9.1 × 10^−1^ and 9.8 × 10^−1^, and 0.809 and 0.875 for the EC and the EC + PEG microparticles, respectively (Table 2).

Determination coefficients of 0.841 and 0.768 were obtained for EC-based microparticles and microparticles composed by the EC + PEG blend, respectively, when the Weibull model was fitted to the RA release profiles (Table 2). The *β* parameter was lower than 1 for both types of microparticles (*β* = 3 × 10^−1^ for EC-based microparticles and *β* = 2 × 10^−1^ for EC + PEG-based microparticles [Table 2]), indicating that the shape of the curve would show a steeper increase regarding the shape of the release curve that corresponds exactly to the shape of an exponential profile [29]. At last, the *τ_d_* parameter was 2 × 10^−1^ and 4 × 10^−4^ min when the polymer EC and the EC + PEG blend were used, respectively (Table 1).

Weibull was the most suitable model to illustrate the release of RA from all polymer-based microparticles proposed in this work. It is explained by the internal morphological structure of spray-dried microparticles [27]. Gonçalves et al. [15,16,17] observed that this model also enabled the best fit to the obtained RA-controlled release profiles from different formulations of biopolymer-based microparticles produced by spray-drying.

## 4. Conclusions

The synthetic polymers EC and PEG were successfully used to produce RA-loaded microparticles by spray-drying. Herein, controlled delivery systems individually composed by EC and composed by the EC + PEG blend were developed. All the formulations enabled obtaining microparticles with a small global size (differential number distribution) and similar surface morphology. Loaded microparticles incorporated the total amount of RA fed to the spray-dryer, obtaining an encapsulation efficiency of around 100%. At last, both EC− and EC + PEG-based microparticles enabled a very fast complete release of RA (in 40 min). Incorporating PEG in the EC-based microparticles did not change the RA release profiles.

## Figures and Tables

**Figure 1 foods-11-02533-f001:**
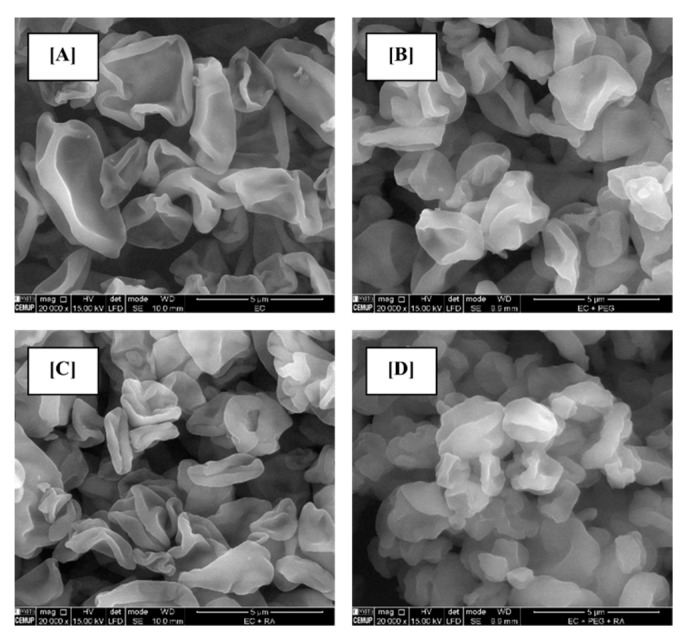
Surface morphology of MpPE (**A**,**B**) and MpPRA (**C**,**D**) microparticles composed by EC (**A**,**C**) and EC + PEG blend (**B**,**D**). Magnification = 20,000 times, beam intensity (HV) 15.00 kV, distance between the sample and the lens (WD) around 10 mm.

**Figure 2 foods-11-02533-f002:**
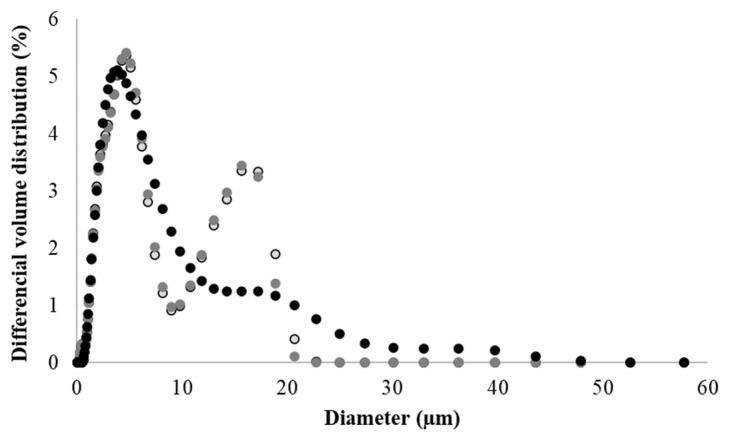
Particle size distribution of microparticles MpPE composed by the EC + PEG blend, considering differential volume distribution: replicate 1—grey circles with black line, replicate 2—dark grey circles and replicate 3—black circles.

**Figure 3 foods-11-02533-f003:**
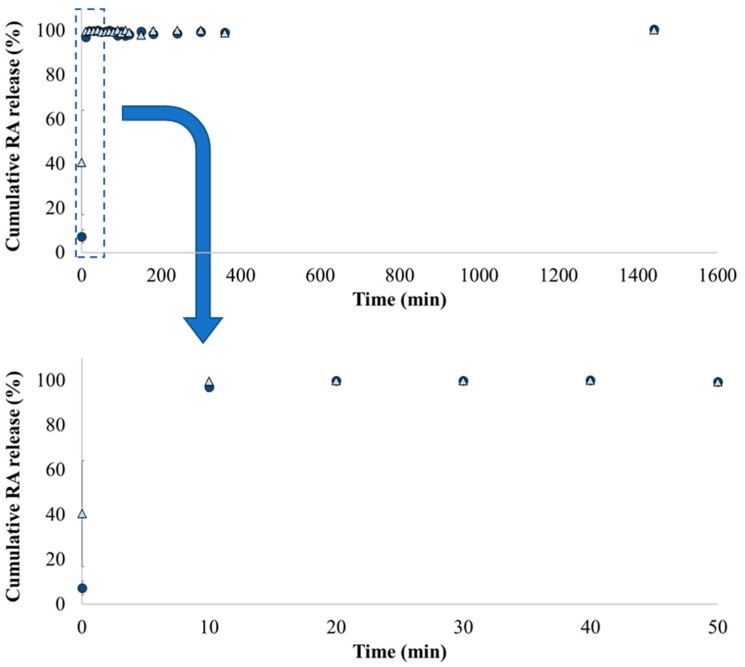
Cumulative release of RA from polymer-microparticles composed by EC (dark blue circles) and by the EC + PEG blend (light blue triangles). Results are presented as the average and standard deviation of three independent assays.

**Table 1 foods-11-02533-t001:** Characterization of polymer-based microparticles for RA encapsulation: product yield, particle diameter, encapsulation efficiency, and loading capacity.

Microparticles Content		Product Yield (%)	Particle Mean Diameter					Encapsulation Efficiency (%)	Loading Capacity (%)
Formulation	Encapsulating Agent		Differential Volume Distribution (µm)	Differential Number Distribution (µm)	D (4,3) (µm)	D (3,2) (µm)	Span (µm)		
**MpPE**	EC	19	22 ± 2 ^bc^	0.090 ± 0.002 ^c^	22 ± 2 ^bc^	1.97 ± 0.03 ^d^	3.5 ± 0.2 ^a^	**-**	**-**
	EC + PEG	21	6.0 ± 0.1 ^c^	0.54 ± 0.02 ^a^	6.0 ± 0.1 ^c^	3.0 ± 0.3 ^c^	3.0 ± 0.3 ^a^	**-**	**-**
**MpPRA**	EC	12	44 ± 8 ^a^	0.13 ± 0.01 ^b^	44 ± 8 ^a^	5.2 ± 0.3 ^b^	6.1 ± 0.5 ^a^	107 ± 23 ^a^	10 ± 2 ^a^
	EC + PEG	28	31 ± 13 ^ab^	0.0987 ± 0.0004 ^c^	31 ± 13 ^ab^	6.1 ± 0.4 ^a^	7 ± 3 ^a^	109 ± 19 ^a^	10 ± 2 ^a^

Different letters in the same column indicate significant differences (*p* < 0.05).

**Table 2 foods-11-02533-t002:** Parameters and correlation coefficients of zero-order, Higuchi, Korsmeyer-Peppas, and Weibull models fitted to the experimental RA release profiles.

Parameters			Encapsulating Agents	
			EC	EC + PEG
**Kinetic models**	**Zero-order**	***K*_0_ (mg∙min^−1^)**	7.1 × 10^−3^ ± 4.6 × 10^−4^	7.2 × 10^−3^ ± 4.7 × 10^−4^
		***Q*_0_ (mg)**	2.2 × 10^−2^ ± 1.4 × 10^−4^	2.4 × 10^−2^ ± 1.6 × 10^−3^
		**R^2^**	0.774	0.751
	**Higuchi**	***K_H_* (mg∙min^−0.5^)**	3.6 × 10^−2^ ± 2.3 × 10^−3^	3.7 × 10^−2^ ± 2.4 × 10^−3^
		**R^2^**	0.977	0.972
	**Korsmeyer-Peppas**	***K_k_* (min^−n^)**	9.1 × 10^−1^ ± 6.0 × 10^−2^	9.8 × 10^−1^ ± 6.4 × 10^−2^
		** *n* **	0.024 ± 0.002	0.0040 ± 0.0003
		**R^2^**	0.809	0.875
	**Weibull**	***τ_d_* (min)**	2 × 10^−1^ ± 1 × 10^−2^	4 × 10^−4^ ± 2.6 × 10^−5^
		** *β* **	3 × 10^−1^ ± 2 × 10^−2^	2 × 10^−1^ ± 1 × 10^−2^
		**R^2^**	0.841	0.768

## Data Availability

Data that support the findings of this study are available upon request to the authors.

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
