# Peer review of "Application of Ethyl Cellulose and Ethyl Cellulose + Polyethylene Glycol for the Development of Polymer-Based Formulations using Spray-Drying Technology for Retinoic Acid Encapsulation"

_foods, 2022, doi:10.3390/foods11162533_

Round 1
Reviewer 1 Report
Ladies and Gentlemen,
The subject is interesting and the introduction explains it comprehensively.
Materials and Methods
There is no information on how many samples were taken for statistical analysis and how many repetitions were made.
Result and Discussion
Figure 2
The chart is not legible. There are no differences between the gray and dark gray circles dots.
Figure 3
The chart is not legible. Markers, error bars and changes that occur within 10-40 minutes are missing. Since after 40 minutes almost 100% of RA has been released, shouldn't samples be taken every 5 minutes instead of 10 minutes?
Line 322 and 323 - no constant units.
Table 2.
The standard errors of the data given in the tables are missing.
Author Response
Response to the reviewers
We would like to thank the comments of the reviewers on the manuscript “Application of ethyl cellulose and ethyl cellulose + polyethylene glycol for the development of polymer‑based formulations by spray-drying technology for retinoic acid encapsulation“, Ref. No. foods-1835591.
The manuscript has been modified according to the suggestions of the reviewers.
Reviewer 1:
Ladies and Gentlemen,
The subject is interesting and the introduction explains it comprehensively.
The authors would like to thank the positive appreciation of this work.
Materials and Methods
There is no information on how many samples were taken for statistical analysis and how many repetitions were made.
Measurement of particle size of microparticles was performed in triplicate. This information is described in section 2.4. The analysis of encapsulation efficiency and loading capacity was also performed in triplicate. This information was included in section 2.5.
Result and Discussion
Figure 2
The chart is not legible. There are no differences between the gray and dark gray circles dots.
Figure 2 was changed.
Figure 3
The chart is not legible. Markers, error bars and changes that occur within 10-40 minutes are missing. Since after 40 minutes almost 100% of RA has been released, shouldn't samples be taken every 5 minutes instead of 10 minutes?
Figure 3 was changed. In this figure, the error bars are too small within 10-40 minutes, whereby it seems they are absent. Considering that RA complete release was achieved in a short period of time, samples could have be taken every 5 min and it is an indication for further experiments. However, samples taken every 10 min showed to be suitable to drawn the RA release profiles from EC and EC+PEG-based microparticles.
Line 322 and 323 - no constant units.
The units were added.
Table 2.
The standard errors of the data given in the tables are missing.
The standard errors were added.
Reviewer 2 Report
In general, the work is well described and written, however, I have some comments:
*. It is suggested to incorporate the characteristics of synthetic polymers – ethyl cellulose (EC) and polyethylene glycol (PEG) – as wall materials for the manufacture of microcapsules (line 73) (i.e. physicochemical, thermal, mechanical properties) and join it with what is written in lines 83 – 87.
*. In line 110 specify the stirring conditions.
*. In line 187 “However, the solution composed by RA 187 and PEG dissolved in absolute ethanol fed to the spray-dryer resulted in microparticles 188 with a crystalline appearance that remained adhered to the cyclone of spray-dryer”, please explain what was due this behaviour.
*. In line 231 “Microparticles MpPE and MpPRA, both composed by EC and by the EC + PEG blend, 231 exhibited an irregular form that varied between a spherical and elliptic structure (Fig-232 ure 1). Moreover, they evidenced several concavities that determined a rough surface. In 233 this case, the production of microparticles with different formulations did not result in 234 different microparticle morphologies”, please explain what was due this behaviour.
*. In the section 3.2., please add: Volume weighted mean diameter D[4,3], area-volume mean diameter D[3,2] and span of the volume-based distribution.
Author Response
Response to the reviewers
We would like to thank the comments of the reviewers on the manuscript “Application of ethyl cellulose and ethyl cellulose + polyethylene glycol for the development of polymer‑based formulations by spray-drying technology for retinoic acid encapsulation“, Ref. No. foods-1835591.
The manuscript has been modified according to the suggestions of the reviewers.
Reviewer 2:
In general, the work is well described and written, however, I have some comments:
The authors would like to thank the global and positive appreciation of the manuscript. The manuscript was reviewed and improved considering the guidelines provided in the following.
*. It is suggested to incorporate the characteristics of synthetic polymers – ethyl cellulose (EC) and polyethylene glycol (PEG) – as wall materials for the manufacture of microcapsules (line 73) (i.e. physicochemical, thermal, mechanical properties) and join it with what is written in lines 83 – 87.
Other characteristics of the synthetic polymers used were added to the manuscript.
*. In line 110 specify the stirring conditions.
The stirring conditions (500 rpm) were added to the manuscript.
*. In line 187 “However, the solution composed by RA 187 and PEG dissolved in absolute ethanol fed to the spray-dryer resulted in microparticles 188 with a crystalline appearance that remained adhered to the cyclone of spray-dryer”, please explain what was due this behaviour.
Briefly this behaviour can be due to the PEG responses to the heating and cooling rate during spray-drying operating conditions. The behaviour of PEG crystallization can change with thermal cycles. This information was added to the manuscript (section 3).
*. In line 231 “Microparticles MpPE and MpPRA, both composed by EC and by the EC + PEG blend, 231 exhibited an irregular form that varied between a spherical and elliptic structure (Fig-232 ure 1). Moreover, they evidenced several concavities that determined a rough surface. In 233 this case, the production of microparticles with different formulations did not result in 234 different microparticle morphologies”, please explain what was due this behaviour.
The polymers used as encapsulating agents can influence the morphology of microparticles produced. Moreover, it can be influenced by the spray-drying process. This information was added to the manuscript (section 3.1).
*. In the section 3.2., please add: Volume weighted mean diameter D[4,3], area-volume mean diameter D[3,2] and span of the volume-based distribution.
Volume weighted mean diameter D[4,3], area-volume mean diameter D[3,2] and span of the volume-based distribution were added to the manuscript (section 3.2.)
Round 2
Reviewer 2 Report
The manuscript has been modified according to my suggestions, it can be accepted for publication in Foods Journal.